# Hyperbranched Polyester Polyfumaratomaleate Doped with Gd(III) and Dy(III) Ions: Synthesis, Structure and Properties

**DOI:** 10.3390/polym14235298

**Published:** 2022-12-04

**Authors:** Aleksei Maksimov, Alina Vagapova, Marianna Kutyreva, Gennadii Kutyrev

**Affiliations:** 1Institute of Polymers, Kazan National Research Technological University, 68 St. K. Marx, 420015 Kazan, Russia; 2A.M. Butlerov Chemistry Institute, Kazan Federal University, 18 Kremlyovskaya Str., 420008 Kazan, Russia

**Keywords:** hyperbranched polyester, metal–polymer complexes, lanthanides

## Abstract

For the first time, metal–polymer complexes have been synthesized using hyperbranched polyester polyfumaratomaleate as a matrix, the structure of which has been established by ^1^H NMR, IR, electron spectroscopy, and elemental analysis methods. The formation of complexes with Gd(III) and Dy(III) ions involving fumarate and maleate groups of the polyester was proved by IR and electron spectroscopy methods. It was established that the structure of the coordination units has the form of a square antiprism. The compositions and conditional logarithms of the stability constants of the complexes were determined. It was established that complexation with lanthanide ions promotes emission enhancement in the ligand.

## 1. Introduction

Compounds of rare-earth elements are effectively used in contrasting reagents for MRI imaging of organ and tissue pathologies [1,2,3]. To date, among the designated in-organic ions, the most significant for practical commercial use are lanthanide complexes, in particular, gadolinium (T1 contrast agents) and dysprosium (T2 contrast agents) [4,5,6]. For the effective use of lanthanide complex compounds for these purposes, it is necessary to ensure their bioavailability and target properties, as well as guarantee their safety and low toxicity to humans [7]. The latter quality is determined by the thermodynamic and kinetic stability of complexes of lanthanide ions [8,9,10,11,12]. However, this problem has not yet been fully resolved. One of the solutions is the search for new ligands for the chelation of lanthanide ions. The use of polymeric ligands for these purposes makes it possible to realize the effect of macromolecular stabilization and obtain more stable complexes [13,14]. The architecture of polymers and the nature of their functional groups are key factors in these developments.

The use of hyperbranched polymers with a 3D architecture and a high density of peripheral functional groups as a platform could be very promising for the creation and stabilization of gadolinium and dysprosium complexes. Hyperbranched polymers are densely packed macromolecules whose three-dimensional framework consists of a core and branching short chains—dendrons. One representative of such structures is hyperbranched polyesters of the Boltorn H series. These polymers consist of ethoxylated pentaerythritol as a core and elementary units formed by polycondensation of 2,2-bis(hydroxymethyl)propionic acid. These nanosized polyesters are industrially available, biodegradable, well-soluble in various polar solvents, and have low toxicity, and the reactive terminal hydroxyl groups can be easily modified to give them the desired set of properties. The introduction of coordinatively active functional groups made it possible to obtain polydentate ligands that form stable complexes with transition metal ions Co(II), Ni(II), and Cu(II) [13]. It has been shown that metal complexes of polyester polycarboxylates form tetrahedral and octahedral [15,16] coordination units and possess marked biological activity [15,17]. The Cu(II) complex based on a third-generation hyperbranched polyester with terminal fumarate groups possesses ferromagnetic properties [16]. At the same time, complexes of hyperbranched polyesters with lanthanide ions are not known [18,19,20].

Since the luminescence mechanism of lanthanide coordination compounds consists of light absorption by a ligand and energy transfer through its triplet level to the lanthanide ion, which emits it as an electromagnetic wave, the molecular design of luminophores is reduced to the search for suitable ligand–lanthanide pairs that form stable coordination compounds and have an optimal ligand triplet level for the excitation of the corresponding lanthanide ion resonance level [21]. Due to the weak complexing ability of lanthanides(III), the formation of stable complexes is possible only for polydentate chelating ligands.

The thermodynamic stability of complexes of lanthanides with derivatives of maleic and fumaric acids has been shown by numerous studies [22,23,24,25,26]. In connection to this, the aim of this work is to synthesize and study the structure and photophysical properties of new complexes of Gd(III) and Dy(III) ions with second-generation hyperbranched polyester with end fumarate and maleate groups.

## 2. Materials and Methods

### 2.1. Materials and Reagents

We used second-generation Boltorn H hyperbranched polyester polyol with 16 terminal hydroxyl groups (Mr = 1750 g mol^−1^, Perstorp Speciality Chemicals AB, Sweden), maleic anhydride (99%, Acros Organics, Waltham, MA, USA), SnCl_2_ (98%, Acros Organics, Waltham, MA, USA), Gd(NO_3_)_3_∙5H_2_O (99.9%, Acros Organics GmbH, Waltham, MA, USA), and Dy(NO_3_)_3_∙5H_2_O (99.9%, Alfa Aesar GmbH, Haverhill, MA, USA). The organic solvents used were 1.4-dioxane, benzene, acetone, ethanol, methanol, and distilled water.

### 2.2. Equipment

^1^H NMR spectra were recorded on a Bruker Avance 400 multifunctional Fourier-transform spectrometer with an operating frequency of 400 MHz in a DMSO-d_6_ solution. IR spectra were recorded on an Infralum FT-08 IR-Fourier spectrometer: resolution 1 cm^−1^, shooting range 4000~400 cm^−1^, KBr pellet. Electronic absorption spectra were recorded on a Lambda 750 spectrophotometer (Perkin-Elmer, Waltham, MA, USA) in the wavelength range of 230~860 nm at *T* = 25 ± 0.01 °C, with an absorbing layer thickness of *l* = 1 cm. Elemental analysis was performed using a Euro EA 3000 CHNS analyzer (Italy). Luminescence spectra were recorded using a fluorescent spectrometer LS 55 (Perkin-Elmer, Waltham, MA, USA) with an excitation wavelength of 200 to 800 nm and an emission of 200 to 900 nm, excitation and emission slit of 5 nm, speed of 100 nm min^−1^ with weak gain at *T =* 25 ± 0.01 °C, and the thickness of the absorbing layer *l =* 1.5 cm. 

The composition and stability of complexes in the lanthanide salt–sodium salt systems of hyperbranched polyester polyfumaratomaleate were determined by electron absorption spectra using Job’s Method [27,28]. The composition and logarithms of the conditional stability constants were calculated using the following parameters for the system: Gd(NO_3_)_3_-compound **2**: *c***_2_** = 4.1 × 10^−4^ − 5.1 × 10^−5^ mol L^−1^, *c*_Gd(NO3)3_ = 5.1 × 10^−4^ − 1.0 × 10^−4^ mol L^−1^, *λ* = 255 nm, *ε*_complex_ = 7648.78 L∙cm^−1^∙mol^−1^, *λ* = 281 nm, *ε*_complex_ = 2715.26 L∙cm^−1^∙mol^−1^, *λ* = 304 nm, *ε*_complex_ = 1132.75 L∙cm^−1^∙mol^−1^; for the system Dy(NO_3_)_3_-compound **2**: *c***_2_** = 4.1 × 10^−4^ − 5.1 × 10^−5^ mol L^−1^, *c*_Dy(NO3)3_ = 5.1 × 10^−4^ − 1.0 × 10^−4^ mol L^−1^, *λ* = 262 nm, *ε*_complex_ = 6423.92 L∙cm^−1^∙mol^−1^, *λ* = 279 nm, *ε*_complex_ = 2997.41 L∙cm^−1^∙mol^−1^, *λ* = 302 nm, *ε*_complex_ = 1202.28 L∙cm^−1^∙mol^−1^.

### 2.3. Synthesis of Compounds

#### 2.3.1. Synthesis of Hyperbranched Polyester with End Fumarate and Maleate Groups (1) 

A 2.50 g (1.43 mmol) sample of second-generation hyperbranched polyester polyol was incubated at 140 °C for 40 min for dehydration and destruction of the self-associates. After cooling to 50 °C, the polymer was dissolved in 10 mL of 1.4-dioxane, then a solution of 1.96 g (19.99 mmol) maleic anhydride and 0.025 g SnCI_2_ (0.132 mmol) as a catalyst in 4 mL of 1.4-dioxane was added to the solution. The reaction was carried out under stirring and at 100 °C for 36 h. The product was precipitated with benzene and dried in a thermal vacuum cabinet at 50 °C and a vacuum of 1 mmHg until constant weight. Compound **1** was a light-yellow resin with a yield of 57%.

^1^H NMR spectrum [(CD_3_)_2_SO], δ ppm: 0.90–1.27 (36 H, CH_3_); 3.29–3.66 (24 H, CH_2_O); 3.95–4.36 (48 H, CH_2_OC(O)); 6.22–6.32 (12 H, CH=CH_cis_); 6.37–6.46 (12 H, CH=CH_trans_). IR spectrum, *ν*, cm^−1^: 3712−3118 (O−H_a._); 3712−3118, 2777, 2587 (O−H_c.a._); 3095, 3064 (C−H_C=C_); 2970, 2923, 2890, 2863 (CH_3_, CH_2_); 1735 (C=O_ester,c.a._); 1640 (C=C_cis_); 1467, 1417, 1377 [δ (CH_3_, CH_2_)]; 1417, 685, 644, 614, 412 [δ (C−H_C=C,cis_)]; 1296−1009 (C−O_ester,ether,a.,c.a._); 918 [δ (O−H_c.a_)]; 894, 868, 820 [δ (C−H_C=C,trans_)]. Elemental analysis. Found, %: C 50.42; H 5.18. C_121_H_148_O_80_. Calculated, %: C 51.11; H 4.89. UV–Vis spectrum (DMSO), *λ*_max_, nm [*ε*, L∙cm^−1^∙mol^−1^]: 260 ROH n→σ^*^, CH=CH π→π^*^, RC(O)OR’ n→π^*^ [3601]; 288 C(O)OH n→π^*^ [614.3].

#### 2.3.2. Synthesis of the Sodium Salt of Hyperbranched Polyester Polyfumaratomaleate (2)

A sample of 1.50 g (0.513 mmol) of compound **1** was dissolved in 10 mL acetone and 0.019 molar NaHCO_3_ in 5 mL ethanol/water (1:1) was added. Product **2** was separated by filtration and dried in a thermal vacuum cabinet at 50 °C and a vacuum of 1 mmHg until constant weight. Compound **2** was a light-yellow powder with a yield of 65%.

IR spectrum, ν, cm^−1^: 3685−3078 (O−H_a._); 3101 (C−H_C=C_); 2955−2855 (CH_3_, CH_2_); 1732 (C=O_ester_); 1593, 1423 (C(O)O^−^); 1469−1358 [δ (CH_3_, CH_2_)]; 1423, 671, 609, 420 [δ (C−H_C=C,cis_)]; 1304−1014 (C−O_ester,ether,a.,c.a._); 879, 833, 817 [δ (C−H_C=C,trans_)]. Elemental analysis. Found, %: C 46.19; H 4.36; Na 8.77. C_121_H_136_Na_12_O_80_. Calculated, %: C 46.55; H 4.05; Na 8.34. UV–Vis spectrum (DMSO), *λ*_max_, nm [*ε*, L∙cm^−1^∙mol^−1^]: 272 ROH n→σ^*^, CH=CH π→π^*^, RC(O)OR’ n→π^*^ [3866]; 305 C(O)O^−^ n→π^*^ [1009].

#### 2.3.3. General Methodology for the Synthesis of Gd(III) (3) and Dy(III) (4) Complexes with Compound (2)

Molar solutions of Gd(NO_3_)_3_∙5H_2_O and Dy(NO_3_)_3_∙5H_2_O in 5 mL of methanol were added to solutions of 0.5 g (0.157 mmol) of product **2** in 2 mL of methanol 0.0022. The isolated complexes were dried in a thermal vacuum oven at 50 °C and a vacuum of 1 mmHg until constant weight. 

The Gd(III) complex **3** is an amorphous substance of light-gold color, 85% yield. IR spectrum, ν, cm^−1^: 3685−3044 (O−H_a._); 3063 (C−H_C=C_); 2959−2854 (CH_3_, CH_2_); 1732 (C=O_ester._); 1647, 774 [δ (NO_3_^−^)]; 1577, 1508−1342 (C(O)O^−^); 1508−1342 [δ (CH_3_, CH_2_)]; 1508−1342, 671, 609, 420 [δ (C−H_C=C,cis_)]; 1300−1006 (C−O_ester,ether,a.,c.a._); 856, 814 [δ (C−H_C=C,trans_)]. Elemental analysis. Found, %: C 38.14; H 4.06; S, 6.46; Gd 15.85. C_126_H_160_Gd_4_O_88_S_8_. Calculated, %: C 37.87; H 4.11; S, 6.13; Gd 16.03. UV–Vis spectrum (DMSO), *λ*_max_, nm: 254 ROH n→σ^*^, CH=CH π→π^*^, RC(O)OR’ n→π^*^; 304 C(O)O^−^ n→π^*^, Gd(III) ^8^S_7/2_→^6^P_7/2_, ^6^P_5/2,_ ^6^I_7/2_, ^6^I_9/2_, ^6^I_17/2_, ^6^I_11/2_, ^6^I_13/2_, ^6^I_15/2_, ^6^D_9/2_. 

The Dy(III) complex **4** is an amorphous compound of light-beige color, 89% yield. IR spectrum, *ν*, cm^−1^: 3692−3036 (O−H_a._); 3105 (C−H_C=C_); 2959−2858 (CH_3_, CH_2_); 1732 (C=O_ester._); 1643, 814, 714 [δ (NO_3_^−^)]; 1585, 1462−1350 (C(O)O^−^); 1462−1350 [δ (CH_3_, CH_2_)]; 1462−1350, 663, 609, 428 [δ (C−H_C=C,cis_)]; 1304−1006 (C−O_ester,ether,a.,c.a._); 852, 814 [δ (C−H_C=C,trans_)]. Elemental analysis. Found, %: C 36.43; H 3.89; S, 11.31; Dy 15.28. C_129_H_164_Gd_4_O_88_S_15_. Calculated, %: C 36.09; H 4.11; S, 11.43; Dy 15.36. UV–Vis spectrum (DMSO), *λ*_max_, nm: 260 ROH n→σ^*^, CH=CH π→π^*^, RC(O)OR’ n→π^*^; 302 C(O)O^−^ n→π^*^; 350 Dy(III) ^6^H_15/2_→^6^P_7_/_2_; 365 Dy(III) ^6^H_15/2_→^4^P_5/2_; 387 Dy(III) H_15/2_→^4^I_13_/_2_; 450 Dy(III) ^6^H_15/2_→^4^I_15_/_2_; 758 Dy(III) ^6^H_15/2_→^6^F_3/2_; 808 Dy(III) ^6^H_15/2_→^6^F_5/2_.

## 3. Results

Hyperbranched polyester polyfumaratomaleate **1** was synthesized by the reaction of second-generation hyperbranched polyester polyol with maleic anhydride (Figure 1A). Product **1** was obtained as a light-yellow resin in a 57% yield. To study the complexing properties of the polymer product with lanthanide ions, at the first stage, sodium salt of hyperbranched polymer **2** was synthesized in a 65% yield (Figure 1B), then complexes with Gd(III) ions **3** and Dy(III) **4** were obtained (Figure 1C).

### 3.1. ^1^H NMR Spectra

When comparing the ^1^H NMR spectra of compound **1** with the original hyperbranched polyester polyol, we observe the appearance of the resonance signal of olefin protons in the fumarate and maleate fragments in the region of 6.21–6.48 ppm (Figure 2) [29].

In the process of attachment of maleic anhydride to the polyester polyol, the cis-configuration of the double bond of the terminal fragments partially transforms into the trans-form. The degree of isomerization of maleate groups (6.32–6.22 ppm) into fumarate groups (6.46–6.37 ppm) in the macromolecule of compound **1** was determined by the ratio of the integral intensities of the resonance signals of protons at the double bond of fumarate groups to the resonance sum of olefin protons, the degree of isomerization being approximately 50% [30].

The degree of functionalization of end hydroxyl groups of hyperbranched polyester polyol was estimated by calculating the ratio of the integral signal intensities of olefin and methyl groups of the polyester complex ester skeleton. It was found that the degree of functionalization is 75%; compound **1** contains six fumarate and six maleate fragments.

### 3.2. IR Spectra

When comparing the IR spectra of the original polyester polyol with compound **1** (Figure 3A), there is a decrease in the intensity of the bands of valent vibrations of hydroxyl groups in the region of 3712−3118 cm^−1^ and an appearance of bands at 2777 and 2587 cm^−1^, due to the presence of bound valent vibrations O−H in the carboxylate fragments [31]. A decrease in the intensity of the C−O bond valence oscillation bands of the original polyester polyol at 1048 and 1009 cm^−1^ is also observed. The appearance of C−H absorption bands of the valence vibrations of the double bond as a duplet at 3096 cm^−1^ and 3064 cm^−1^ confirms the presence of fumarate and maleate groups [32]. Cis-trans-isomerization is also confirmed by the appearance of triplet peaks of the strain vibrations of C−H_C=C_ bonds at 894, 868, 820 cm^−1^ (tras-isomer) and 685, 644, 614 cm^−1^ (cis-isomer). An approximation of the absorption bands of compound **1** at 1735 and 1640 cm^−1^ by the Gaussian method showed the presence of bands of valent vibrations of C=O bonds in the ester and acidic fragments at 1738 and 1695 cm^−1^, and the bands at 1648 and 1632 cm^−1^ correspond to valent vibrations of C=C bonds in the fumarate and maleate groups (Figure 3B) [32,33].

In the IR spectrum of compound **2**, the presence of valence bands of asymmetric vibrations of carboxylate anion groups of fumarate and maleate fragments at 1599 and 1564 cm^−1^ (Figure 4B) was found [34]. After an approximation of the absorption bands by the Gaussian method in the region from 1675 to 1375 cm^−1^, the presence of a band of valence vibrations of the C=C bond at 1641 cm^−1^ in the maleate and fumarate groups was observed [32,33]. The band at 1461 cm^−1^ corresponds to valence symmetric vibrations in the carboxylate anion group, and the band at 1423 cm^−1^ is attributed to strain vibrations of the C−H_C=C_ bond in the maleate fragment [34].

A comparison of the IR spectra of the Gd(III) complex (3) with compound (2) (Figure 5) demonstrated an increase in the intensity of the C=O stretching vibration bands in the ester group at 1732 cm^−1^. In addition, there is a decrease in the intensity of the peaks of the stretching vibrations of C(O)O– and C=O bonds at 1577, 1508–1342 and 1041, 1006 cm^−1^. In addition, when comparing the IR spectra of the Dy(III) complex (3) with compound (2), an increase in the intensity of the bands of the stretching vibrations of the C=O bond in the ester group at 1732 cm^−1^ and a decrease in the intensity of the bands of the stretching vibrations of the C(O)O– and C=O bonds at 1585, 1462–1350 and 1304–1006 cm^−1^.

All these facts suggest that the Gd(III) and Dy(III) ions in complexes **3** and **4** are in coordination with the oxygen atoms of the carboxylate anionic and ester groups. In addition, the coordination sites of complexes **3** and **4** contain nitro groups at 1647, 774, and 1643, 814, 714 cm^−1^.

### 3.3. Electronic Spectra

Hyperbranched polyester polyfumaratomaleate **1** and its complexes with Gd(III) and Dy(III) ions were studied by spectrophotometry. The obtained plots were subjected to interpolation by the Gaussian distribution function, and the results of the analysis are presented in Figure 6 and Table 1.

An examination of the electronic spectrum of compound **1** solution in DMSO (Figure 6A) revealed the presence of absorption bands at 259 and 263 nm, referred to as π→π^*^ transitions in the olefin fragments of fumarate and maleate groups. The absorption band at 288 nm corresponds to n→π^*^ transitions in the carboxylate group [35]. The interpolation of the absorption bands of compound **2** solution in DMSO (Figure 6B) showed the disappearance of the absorption band at 288 nm and the appearance of bands at 282 and 305 nm related to n→π^*^ transitions in carboxylate anion fragments of fumarate and maleate groups [35].

The interpolation of the absorption bands of the complexes with Gd(III) ions **3** (Figure 6C) and Dy(III) **4** (Figure 6D,E) in DMSO solutions showed that all the absorption bands present in the spectrum of compound **2** were preserved. The absorption band at 304 nm in compound **3** is attributed to f→f transitions in the Gd(III) ion from the ^8^S_7/2_ level to the ^6^P_7/2_, ^6^P_5/2,_^6^ I_7/2_, ^6^I_9/2_ ^6^I_17/2_, ^6^I_11/2_, ^6^I_13/2_, ^6^I_15/2_, ^6^D_9/2_ levels, indicating the presence of eight coordinated Gd(III) ions in the DMSO solution (Figure 6C) [36]. In the spectrum of the Dy(III) complex **4**, the appearance of bands at 350, 365, 387, 450, 758, and 808 nm was observed in the visible region (Figure 6E), which correspond to transitions between the ground state ^6^H_5/2_ and multiple excited states belonging to the configuration 4f^9^ of the ion Dy(III) from ^6^H_15/2_ to ^6^P_7/2_, ^4^P_5/2_, ^4^I_13/2_, ^4^I_15/2_, ^6^F_3/2_ and ^6^F_5/2_, respectively [37,38,39,40,41,42]. It was found that complex **4** with Dy(III) ions has a coordination node in the form of a square antiprism [43,44].

**Table 1 polymers-14-05298-t001:** Deconvolution of electronic spectra of compounds **1**, **2** and complexes Gd(III) **3,** Dy(III) **4** in DMSO.

(1) λ, nm	(2) λ, nm	(3) λ, nm	(4) λ, nm	Transition	Band, Bond and Metal	Literature
255	256	255	255	n→σ^*^	ROH	[35]
259	259	258	258	π→π^*^	CH=CH (cis)	[35]
263	264	263	263	π→π^*^	CH=CH (trans)	[35]
272	272	269	269	n→π^*^	RC(O)OR’	[35]
-	282	281	279	n→π^*^	C(O)O^−^ (cis)	[45,46]
288	-	-	-	n→π^*^	C(O)OH	[47]
-	305	304	302	n→π^*^	C(O)O^−^ (trans)	[45,46]
-	-	304	-	^8^S_7/2_→^6^P_7/2_, ^6^P_5/2,_ ^6^I_7/2_, ^6^I_9/2_, ^6^I_17/2_, ^6^I_11/2_, ^6^I_13/2_, ^6^I_15/2_, ^6^D_9/2_,	Gd(III)	[36]
-	-	-	305, 365, 387, 450, 758, 808	^6^H_15/2_→^6^P_7_/_2_, ^4^P_5/2_, ^4^I_13_/_2_, ^4^I_15_/_2_; ^6^F_3/2_; ^6^F_5/2_	Dy(III)	[43,44]

The electronic absorption spectra of the system [Gd(III) and Dy(III) salts—compound **2**] were studied (Figure 7). The band at 262 nm (4.27 A) has a hypsochrome effect with a hypsochrome shift up to 253 nm (0.11 A for compound **3** and 0.04 A for compound **4** when the molar ratios of Gd(III)/Dy(III):compound **2** increase from 0.25:1 to 9:1 (Figure 7A,C)). The band at 304 nm (0.59 A) of compound **3** (Figure 7A) and the band at 302 nm (0.62 A) of compound **4** (Figure 7C) have only a hypsochrome effect to an optical density of 0.03 A and 0.02 A, respectively. All this indicates the participation of carboxylate anions in the coordination with lanthanide ions in the whole range of molar ratios studied. The overall spectral pattern of the [lanthanide(III)-compound **2**] system is similar to the position of the absorption bands in the individual complexes **3** and **4**, indicating the identity of the coordination unit geometry in the condensed phase and in solution, and confirming the realization of the coordination unit geometry with Gd(III) and Dy(III) ions of the square anti-prism type [48,49].

It was found that, in the systems [Gd/Dy—compound **2**] in solution, the complex form of M:L = 4:1 is formed, that is, there is one lanthanide ion per three fumaratomaleate groups of the polymer, which is confirmed by the elemental analysis data (Figure 7B,D). The logarithms of the conditional stability constants for Gd(III) **3** and Dy(III) **4** complexes were *lgβ* = 4.59 ± 0.26 and 4.84 ± 0.07, which indicates the high stability of the complexes with Gd(III) and Dy(III) ions based on hyperbranched polyester polyfumaratomaleate [50].

### 3.4. Luminessence

The emission of hyperbranched polyester polyfumaratomaleate **1** and its complexes with Gd(III) **3** and Dy(III) ions **4** was studied in DMF solution at room temperature (1 × 10^−4^ mol∙L^−1^). Electron spectroscopy showed that the most electrons are absorbed in the 254–260 nm region. However, the maximum absorbed is observed at excitation λ = 235 ± 2 nm, which is due to the high polarity of the solvent used (Figure 8A) [51]. The emission of complexes (3) and (4) is observed in the range from 550 to 650 nm in the visible part of the spectrum (Figure 8B). The peak at 583 nm in the spectrum of complex (4) is attributed to the ^4^F_9/2_→^6^H_15/2_ Dy(III) transition; the emitting light of the complex is yellow–green [52,53]. The peak at 618 nm for complex **3** is attributed to the energy transition from the T1 level of the ligand to the main singlet level (S_0_).

## 4. Conclusions

The reaction of the addition of maleic anhydride to hyperbranched second-generation polyester polyol obtained hyperbranched polyester polyfumaratomaleate **1** in a 57% yield. The structure of compound **1** was proved by ^1^H NMR, IR and electron spectroscopy, and the presence of six fumarate and six maleate fragments at terminal positions was established. The hyperbranched terminally decorated polyester (1) was successfully used as a polydentate ligand for doping with Gd(III) and Dy(III) rare-earth cations.

New complexes with Gd(III) **3** and Dy(III) **4** ions were synthesized based on ligand **1**. The oxygen atoms of fumarate and maleate groups participate in complexation with Gd(III) and Dy(III) ions by IR and electron spectroscopy. According to UV–Vis spectrophotometry, three carboxyl groups of the terminal fumarate or maleate fragments are involved in the coordination of one metal ion. The Gd(III) and Dy(III) coordination sites have the composition [MO6X2] (M= Gd(III), Dy(III), X= NO_3_^−^ or H_2_O_coordination_) and have a quadratic-antiprismatic geometry. The logarithms of the stability constants are *lgβ* = 4.59 ± 0.26 and 4.84 ± 0.07, respectively. The complexation with Gd(III) and Dy(III) ions promotes the luminescence enhancement in comparison with ligand **1** by 16.77 and 202.08 times. The significant difference in the emission intensity of the complex with Dy(III) ions **4** as compared with complex with Gd(III) ions **3** (approximately 18 times) appears to be due to the fact that, in Gd(III), in the first excited state ^6^P_7/2_, Gd(III) is too far away to accept the energy on the triplet excited state of the ligand [52,53]. The studies will be continued to clarify the possibility of using these complexes as contrast agents in MRI.

## Figures and Tables

**Figure 1 polymers-14-05298-f001:**
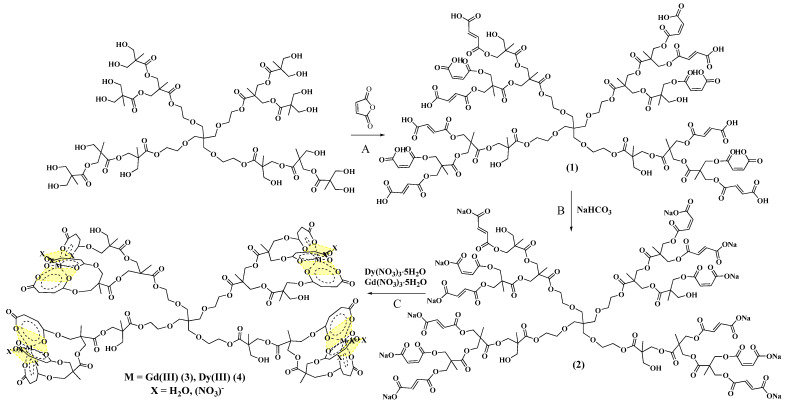
Synthesis of hyperbranched polyester polyfumaratomaleate **1**, salt **2** and complexes with Gd(III) **3** and Dy(III) **4** ions (**A**-the process of adding maleic anhydride to a polyester polyol and obtaining polyfumaratomaleate (**1**), **B**-stage of synthesis of sodium salt of hyperbranched polyfumaratomaleate (**2**), **C**-stage of synthesis of hyperbranched polyfumaratomaleate complexes with Gd(III) (**3**) and Dy(III) (**4**) ions).

**Figure 2 polymers-14-05298-f002:**
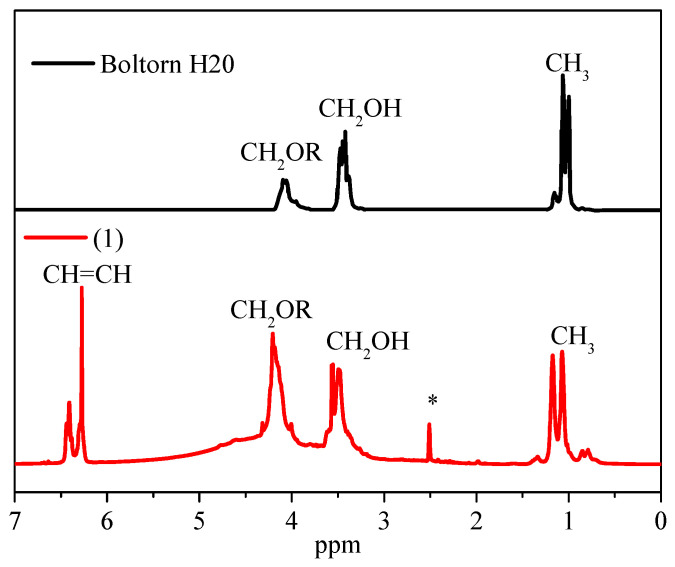
^1^H NMR spectra of the original hyperbranched polyester polyol (Boltorn H20) and compound **1**; *—DMSO proton resonance.

**Figure 3 polymers-14-05298-f003:**
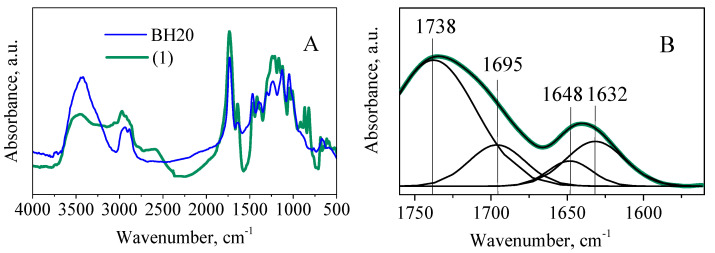
Infrared spectra of Boltorn H20 and compound **1** (**A**) and approximation of absorption bands of compound **1** in the region of 1760 and 1535 cm^−1^ (**B**).

**Figure 4 polymers-14-05298-f004:**
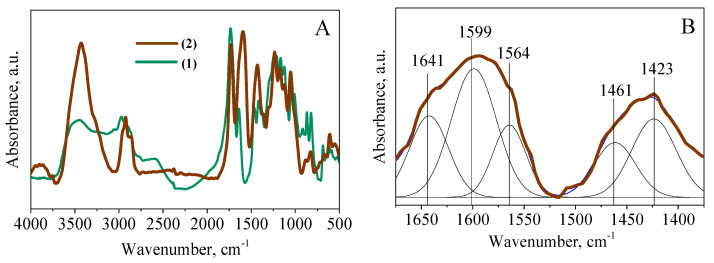
Infrared spectra of compounds **1** and **2** (**A**) and approximation of absorption bands of compound **2** in the region from 1675 to 1375 cm^−1^ (**B**).

**Figure 5 polymers-14-05298-f005:**
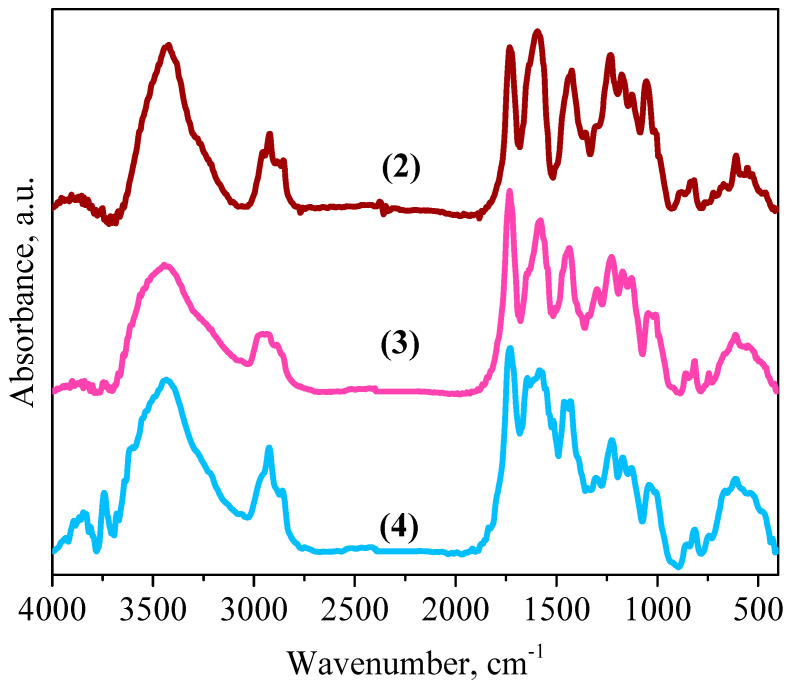
IR spectra of sodium salt **2** and Gd(III) **3** and Dy(III) **4** complexes.

**Figure 6 polymers-14-05298-f006:**
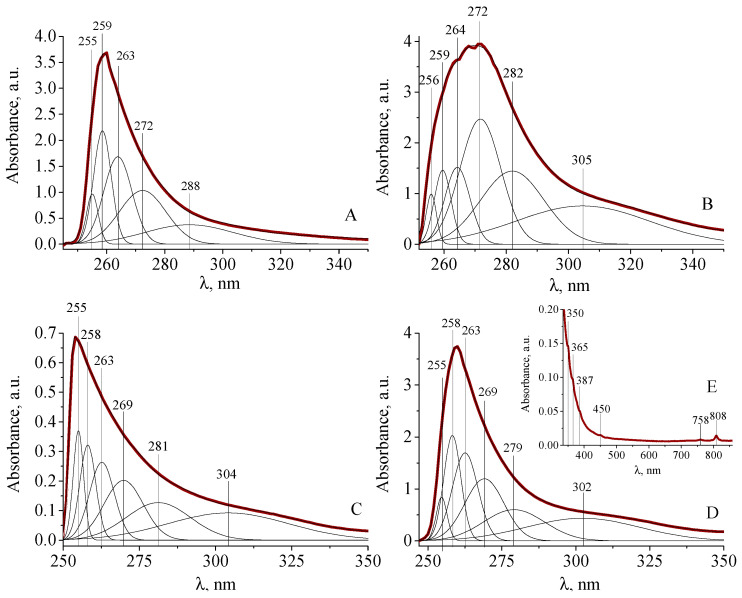
Electronic absorption spectra of compounds **1** and **2**, interpolated from the Gaussian distribution function in the wavelength regions 250–350 (**A**) and 245–350 nm (**B**) in DMSO, *c***_1,2_** = 1 × 10^−3^ mol∙L^−1^; Gd(III) **3** and Dy(III) complexes **4**, interpolated from the Gaussian distribution function in the wavelength regions 250–350, *c***_3_** = 5.0 × 10^−4^ (**C**) and 247–350 nm, *c***_4_** = 5.0 × 10^−3^ mol∙L^−1^ (**D**), in the region from 335 nm to 860 nm (**E**) in DMSO.

**Figure 7 polymers-14-05298-f007:**
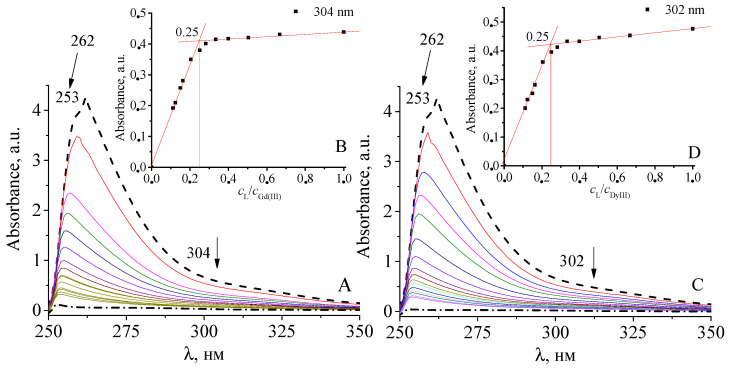
Electronic absorption spectra of compound **2** (dashed line), Gd(NO_3_)_3_∙5H_2_O and Dy(NO_3_)_3_∙5H_2_O (dashed and dotted line), [Gd(NO_3_)_3_/Dy(NO_3_)_3_—compound **2**] (solid line) in the wavelength region 250–350 nm, *c***_2_** = 4.1 × 10^−4^–5.1 × 10^−5^ mol∙L^−1^, *c*_M(NO3)3_ = 5.1 × 10^−4^–1.0 × 10^−4^ mol∙L^−1^ [M = Gy(III), Dy(III)] (**A**,**C**); Dependence of the optical density of the system [Gd(NO_3_)_3_/Dy(NO_3_)_3_—combination **2]** at λ= 304 and 302 nm in DMSO on the ratio *c*_L_/*c*_Gd(III)/Dy(III)_ (**B**,**D**).

**Figure 8 polymers-14-05298-f008:**
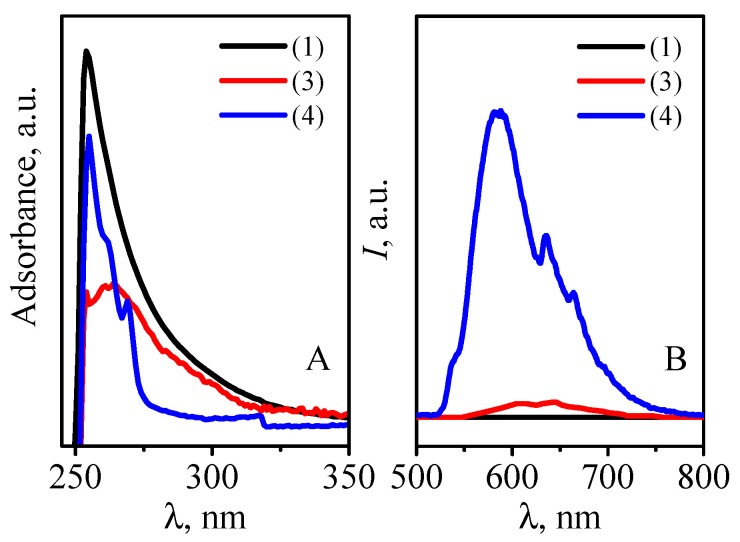
Excitation (**A**) and emission (**B**) spectra of the ligand **1** and Gd(III) **3**, Dy(III) **4** complexes at room temperature, *c***_1,3,4_** = 1.0 × 10^−4^ mol∙L^−1^.

## Data Availability

Not applicable.

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
