# Peer review of "Hyperbranched Polyester Polyfumaratomaleate Doped with Gd(III) and Dy(III) Ions: Synthesis, Structure and Properties"

_polymers, 2022, doi:10.3390/polym14235298_

Round 1
Reviewer 1 Report
The paper is devoted for hyperbranched polyester polyfumaratomaleate doped doped with
Gd and Dy ions synthesis and characterization. The topic is generally interesting, however the paper contains unexplained places (below)
and need major revisions.
The aim of the paper is not clear. Please formulate clearly the aim of the paper.
Figures 3-5, why infrared investigations were performed only in frequency range from
cm-1 to 500 cm-1?
Figure 6 numbers on Y axis should written be separating integer part of number from
fractional by point, for example "4.0", not "4,0".
Results obtained in Fig. 8 should be compared with results presented in literature.
Please rewrite conclusions in more informative way.
Author Response
Dear Editor,
The authors are thank the Reviewers for comments. Appropriate corrections have been made to the article.
The authors draw the editor's attention to the corrections made in figure 1 in accordance with the information in the "Results" section. All changes made to the manuscript in accordance with the comments of the reviewers are marked in green.
Reviewer 1
The paper is devoted for hyperbranched polyester polyfumaratomaleate doped doped with Gd and Dy ions synthesis and characterization. The topic is generally interesting, however the paper contains unexplained places (below) and need major revisions.
1.The aim of the paper is not clear. Please formulate clearly the aim of the paper
Authors comments:
The authors fully agree with this comment and have made the following corrections
Compounds of rare-earth elements are effectively used in contrasting reagents for MRI imaging of organ and tissue pathologies [1-3]. To date, among the designated in-organic ions, the most significant for practical commercial use are lanthanide com-plexes, in particular gadolinium (T1 contrast agents) and dysprosium (T2 contrast agents) [4–6]. For the effective use of lanthanide complex compounds for these purposes, it is necessary to ensure their bioavailability and target properties, as well as guarantee their safety and low toxicity to humans [7]. The latter quality is determined by the thermo-dynamic and kinetic stability of complexes of lanthanide ions [8–12]. However, this problem has not yet been fully resolved. One of the solutions is the search for new ligands for chelation of lanthanide ions. The use of polymeric ligands for these purposes makes it possible to realize the effect of macromolecular stabilization and obtain more stable complexes [13, 14]. The architecture of polymers and the nature of their functional groups are key factors in these developments.
- Alzola-Aldamizetxebarria, S.; Fernández-Méndez, L.; Padro, D.; Ruíz-Cabello, J.; Ramos-Cabrer, P. A Comprehensive Introduction to Magnetic Resonance Imaging Relaxometry and Contrast Agents. ACS Omega 2022, 7, 36905–36917. https://doi.org/10.1021/acsomega.2c03549.
- Avasthi, A.; Caro, C.; Pozo‑Torres, E.; Leal, M.P.; García‑Martín, M.L. Magnetic Nanoparticles as MRI Contrast Agents. In Surface-modified Nanobiomaterials for Electrochemical and Biomedicine Applications; Puente-Santiago, A.R., Rodríguez-Padrón, D., Eds.; Springer International Publishing: Cham, 2020; pp. 49–91 ISBN 978-3-030-55502-3.
- Wahsner, J.; Gale, E.M.; Rodríguez-Rodríguez, A.; Caravan, P. Chemistry of MRI Contrast Agents: Current Challenges and New Frontiers. Rev. 2019, 119, 957–1057. https://doi.org/10.1021/acs.chemrev.8b00363.
- Zapolotsky, E.N.; Qu, Y.; Babailov, S.P. Lanthanide Complexes with Polyaminopolycarboxylates as Prospective NMR/MRI Diagnostic Probes: Peculiarities of Molecular Structure, Dynamics and Paramagnetic Properties. Incl. Phenom. Macrocycl. Chem. 2022, 102, 1–33. https://doi.org/10.1007/s10847-021-01112-3.
- Yue, H.; Park, J.Y.; Chang, Y.; Lee, G.H. Ultrasmall Europium, Gadolinium, and Dysprosium Oxide Nanoparticles: Polyol Synthesis, Properties, and Biomedical Imaging Applications. Mini Rev. Med. Chem. 2020, 20, 1767–1780. https://doi.org/10.2174/1389557520666200604163452.
- Kim, H.-K.; Lee, G.H.; Chang, Y. Gadolinium as an MRI Contrast Agent. Future Med. Chem. 2018, 10, 639–661. https://doi.org/10.4155/fmc-2017-0215.
- Clough, T.J.; Jiang, L.; Wong, K.-L.; Long, N.J. Ligand Design Strategies to Increase Stability of Gadolinium-Based Magnetic Resonance Imaging Contrast Agents. Commun. 2019, 10, 1420. https://doi.org/10.1038/s41467-019-09342-3.
- Garda, Z.; Nagy, V.; Rodríguez-Rodríguez, A.; Pujales-Paradela, R.; Patinec, V.; Angelovski, G.; Tóth, É.; Kálmán, F.K.; Esteban-Gómez, D.; Tripier, R.; et al. Unexpected Trends in the Stability and Dissociation Kinetics of Lanthanide(III) Complexes with Cyclen-Based Ligands across the Lanthanide Series. Chem. 2020, 59, 8184–8195. https://doi.org/10.1021/acs.inorgchem.0c00520.
- Layne, K.A.; Dargan, P.I.; Archer, J.R.H.; Wood, D.M. Gadolinium Deposition and the Potential for Toxicological Sequelae – A Literature Review of Issues Surrounding Gadolinium-Based Contrast Agents. J. Clin. Pharmacol. 2018, 84, 2522–2534. https://doi.org/10.1111/bcp.13718.
- Hu, A.; Keresztes, I.; MacMillan, S.N.; Yang, Y.; Ding, E.; Zipfel, W.R.; DiStasio, R.A.Jr.; Babich, J.W.; Wilson, J.J. Oxyaapa: A Picolinate-Based Ligand with Five Oxygen Donors That Strongly Chelates Lanthanides. Chem. 2020, 59, 5116–5132. https://doi.org/10.1021/acs.inorgchem.0c00372.
- Blomqvist, L.; Nordberg, G.F.; Nurchi, V.M.; Aaseth, J.O. Gadolinium in Medical Imaging–Usefulness, Toxic Reactions and Possible Countermeasures–A Review. Biomolecules 2022, 12. https://doi.org/10.3390/biom12060742.
- Sembo-Backonly, B.S.; Estour, F.; Gouhier, G. Cyclodextrins: Promising Scaffolds for MRI Contrast Agents. RSC Adv. 2021, 11, 29762–29785. https://doi.org/10.1039/D1RA04084G.
- Maksimov, A.; Kutyrev, G. Functionalized Hyperbranched Aliphatic Polyester Polyols: Synthesis, Properties and Applications. Chinese J. Polym. Sci. 2022, 1–19. https://doi.org/10.1007/s10118-022-2823-0.
- Janicki, R.; Mondry, A.; Starynowicz, P. Carboxylates of Rare Earth Elements. Chem. Rev. 2017, 340, 98–133. https://doi.org/10.1016/j.ccr.2016.12.001.
In this work, the authors propose to use hyperbranched polymers with 3D archi-tecture and high density of peripheral functional groups as a polymer platform for the creation and stabilization of gadolinium and dysprosium complexes. Boltorn H series hyperbranched polyester with terminal OH groups was chosen as the object of study. The use of the polyether functionalization approach by fragments of unsaturated carboxylic acids will increase the specificity of complexation of the hyperbranched macroligand with Gd(III) and Dy(III) ions and obtain stable complexes with good target characteristics.
In this manuscript, the authors present the intermediate results of research in project on developing approaches to the synthesis, studying the structure and photophysical properties of new Gd(III) and Dy(III) ions with hyperbranched polymers with end-fumarate and maleate groups.
In this connection, the aim of this work is to synthesize, study the structure and photophysical properties of a new complexes Gd(III) and Dy(III) ions with second-generation hyperbranched polyester with end - fumarate and maleate groups.
Appropriate сorrections have been made to the "Introduction" section (in green).
- Figures 3-5, why infrared investigations were performed only in frequency range from cm-1 to 500 cm-1?
Authors comments:
All FT-IR spectra of all compounds were recorded in the range from 4000 to 400 cm–1 (according to the information in section “2. Materials and Methods/ 2.2. Equipment”). However, in the “Results” section of Figures 3A and 4A, the range from 500 to 400 cm-1 is not presented, since the characteristic bands of compounds (1) and (2) are absent in this range. At the same time, the spectra of compounds (3) and (4) in Figure 5 are presented completely in the range from 4000 to 400 cm–1.
- Figure 6 numbers on Y axis should written be separating integer part of number from fractional by point, for example "4.0", not "4,0".
Authors comments:
Corrections have been made to the entry of numbers in figure 6 on Y axis.
- Results obtained in Fig. 8 should be compared with results presented in literature.
Authors comments:
The authors believe that a quantitative comparison of the photophysical properties of new complexes of Gd(III) (3) and Dy(III) (4) ions with hyperbranched polyester polyfumaratomaleate (1) with the emission of compounds presented in the literature is incorrect. This is due, first of all, to a significant difference in the structure of the organic compounds used in our work. As is known, the structure of the ligand largely determines the photoactivity of the complex, due to the local environment of the metal ion (geometry of the polyhedron, coordination of functional groups that can enhance the luminescence of the metal ion, etc.), the mutual arrangement of metal centers in the macromolecule (the presence or minimization of intermolecular interactions, packing effects). In addition, comparison is possible only under identical experimental conditions.
However, the authors made an attempt to explain the observed photoactivity in section “3.4. Luminessence ". For this, the text of the article is supplemented with references to sources [52-53], where the emission of similar systems was studied.
- Zhang, L.; Ji, Y.; Xu, X.; Liu, Z.; Tang, J. Synthesis, Structure and Luminescence Properties of a Series of Dinuclear LnIII Com-plexes (Ln=Gd, Tb, Dy, Ho, Er). J Lumin 2012, 132, 1906–1909. https://doi.org/10.1016/j.jlumin.2012.03.040.
- Bazhina, E.S.; Bovkunova, A.A.; Medved’ko, A. v; Varaksina, E.A.; Taidakov, I. v; Efimov, N.N.; Kiskin, M.A.; Eremenko, I.L. Lanthanide(III) (Eu, Gd, Tb, Dy) Complexes Derived from 4-(Pyridin-2-Yl)Methyleneamino-1,2,4-Triazole: Crystal Structure, Magnetic Properties, and Photoluminescence. Chem Asian J. 2018, 13, 2060–2068. https://doi.org/10.1002/asia.201800511.Please rewrite conclusions in more informative way.
- Please rewrite conclusions in more informative way.
Authors comments:
The authors took into account the wishes of the reviewer. Conclusions added.
«The reaction of addition of maleic anhydride to hyperbranched second generation polyester polyol obtained hyperbranched polyester polyfumaratomaleate (1) in 57% yield. The structure of compound (1) was proved by 1H NMR, IR and electron spectroscopy and the presence of 6 fumarate and 6 maleate fragments at terminal positions was established. The hyperbranched terminally decorated polyester (1) was successfully used as a polydentate ligand for doping with Gd(III) and Dy(III) rare earth cations.
A new complexes with Gd(III) (3) and Dy(III) (4) ions were synthesized based on ligand (1). The oxygen atoms of fumarate and maleate groups of hyperbranched polymer participate in complexation with Gd(III) and Dy(III) ions by IR and electron spectroscopy. According to UV-Vis spectrophotometry, three carboxyl groups of the terminal fumarate or maleate fragments are involved in the coordination of one metal ion. The Gd(III) and Dy(III) coordination sites have the composition [MO6X2] (M= Gd(III), Dy(III), X= NO3- or H2Ocoordination) and have a quadratic-antiprismatic geometry. Logarithms of the stability constants are lgβ = 4.59 ± 0.26 and 4.84 ± 0.07, respectively.
The complexation with Gd(III) and Dy(III) ions promotes the luminescence enhancement in comparison with ligand (1) by 16.77 and 202.08 times. The significant difference in the emission intensity of the complex with Dy(III) ions (4) as compared with complex with Gd(III) ions (3) (approximately 18-times) appears to be due to the fact that in Gd(III) the first excited state 6P7/2 Gd(III) is too far away to accept the energy on the triplet excited state of the ligand [52,53]. The studies will be continued to clarify the possibility of using these complexes as contrast agents in MRI.»
All comments of the Reviewer are taken into account. Appropriate changes have been made to the manuscript. The authors thank the Reviewer for a detailed review of the work. The comments made are very helpful in moving our project forward in the future.

Reviewer 2 Report
This manuscript deals with the synthesis of hyperbranched polyester bearing fumarate and maleate end groups via reaction of commercial Boltorn H hyperbranched polyester polyol with maleic anhydride followed by the preparation of its complexes with Gd(III) and Dy(III) ions. This paper is well-written, while all conclusions supported by the experimental data. However, the aim of this work is not very clear especially in a view of the further application of the prepared complexes. Therefore, I suggest to authors explaining why they selected Gd(III) and Dy (III) salts among other lanthanides? What is the possible application of such complexes?
In addition, authors reported much higher luminescence enhancement for complex of hyberbranched polyester with Dy (III) as compared to complex with Gd (III) ? What is the reason for that?
Author Response
Dear Editor,
The authors are thank the Reviewers for comments. Appropriate corrections have been made to the article.
The authors draw the editor's attention to the corrections made in figure 1 in accordance with the information in the "Results" section. All changes made to the manuscript in accordance with the comments of the reviewers are marked in green.
Reviewer 2
This manuscript deals with the synthesis of hyperbranched polyester bearing fumarate and maleate end groups via reaction of commercial Boltorn H hyperbranched polyester polyol with maleic anhydride followed by the preparation of its complexes with Gd(III) and Dy(III) ions. This paper is well-written, while all conclusions supported by the experimental data.
1.However, the aim of this work is not very clear especially in a view of the further application of the prepared complexes. Therefore, I suggest to authors explaining why they selected Gd(III) and Dy (III) salts among other lanthanides? What is the possible application of such complexes?
Authors comments:
The authors fully agree with this comment and have made the following corrections
Compounds of rare-earth elements are effectively used in contrasting reagents for MRI imaging of organ and tissue pathologies [1-3]. To date, among the designated in-organic ions, the most significant for practical commercial use are lanthanide com-plexes, in particular gadolinium (T1 contrast agents) and dysprosium (T2 contrast agents) [4–6]. For the effective use of lanthanide complex compounds for these purposes, it is necessary to ensure their bioavailability and target properties, as well as guarantee their safety and low toxicity to humans [7]. The latter quality is determined by the thermo-dynamic and kinetic stability of complexes of lanthanide ions [8–12]. However, this problem has not yet been fully resolved. One of the solutions is the search for new ligands for chelation of lanthanide ions. The use of polymeric ligands for these purposes makes it possible to realize the effect of macromolecular stabilization and obtain more stable complexes [13, 14]. The architecture of polymers and the nature of their functional groups are key factors in these developments.
- Alzola-Aldamizetxebarria, S.; Fernández-Méndez, L.; Padro, D.; Ruíz-Cabello, J.; Ramos-Cabrer, P. A Comprehensive Introduction to Magnetic Resonance Imaging Relaxometry and Contrast Agents. ACS Omega 2022, 7, 36905–36917. https://doi.org/10.1021/acsomega.2c03549.
- Avasthi, A.; Caro, C.; Pozo‑Torres, E.; Leal, M.P.; García‑Martín, M.L. Magnetic Nanoparticles as MRI Contrast Agents. In Surface-modified Nanobiomaterials for Electrochemical and Biomedicine Applications; Puente-Santiago, A.R., Rodríguez-Padrón, D., Eds.; Springer International Publishing: Cham, 2020; pp. 49–91 ISBN 978-3-030-55502-3.
- Wahsner, J.; Gale, E.M.; Rodríguez-Rodríguez, A.; Caravan, P. Chemistry of MRI Contrast Agents: Current Challenges and New Frontiers. Rev. 2019, 119, 957–1057. https://doi.org/10.1021/acs.chemrev.8b00363.
- Zapolotsky, E.N.; Qu, Y.; Babailov, S.P. Lanthanide Complexes with Polyaminopolycarboxylates as Prospective NMR/MRI Diagnostic Probes: Peculiarities of Molecular Structure, Dynamics and Paramagnetic Properties. Incl. Phenom. Macrocycl. Chem. 2022, 102, 1–33. https://doi.org/10.1007/s10847-021-01112-3.
- Yue, H.; Park, J.Y.; Chang, Y.; Lee, G.H. Ultrasmall Europium, Gadolinium, and Dysprosium Oxide Nanoparticles: Polyol Synthesis, Properties, and Biomedical Imaging Applications. Mini Rev. Med. Chem. 2020, 20, 1767–1780. https://doi.org/10.2174/1389557520666200604163452.
- Kim, H.-K.; Lee, G.H.; Chang, Y. Gadolinium as an MRI Contrast Agent. Future Med. Chem. 2018, 10, 639–661. https://doi.org/10.4155/fmc-2017-0215.
- Clough, T.J.; Jiang, L.; Wong, K.-L.; Long, N.J. Ligand Design Strategies to Increase Stability of Gadolinium-Based Magnetic Resonance Imaging Contrast Agents. Commun. 2019, 10, 1420. https://doi.org/10.1038/s41467-019-09342-3.
- Garda, Z.; Nagy, V.; Rodríguez-Rodríguez, A.; Pujales-Paradela, R.; Patinec, V.; Angelovski, G.; Tóth, É.; Kálmán, F.K.; Esteban-Gómez, D.; Tripier, R.; et al. Unexpected Trends in the Stability and Dissociation Kinetics of Lanthanide(III) Complexes with Cyclen-Based Ligands across the Lanthanide Series. Chem. 2020, 59, 8184–8195. https://doi.org/10.1021/acs.inorgchem.0c00520.
- Layne, K.A.; Dargan, P.I.; Archer, J.R.H.; Wood, D.M. Gadolinium Deposition and the Potential for Toxicological Sequelae – A Literature Review of Issues Surrounding Gadolinium-Based Contrast Agents. J. Clin. Pharmacol. 2018, 84, 2522–2534. https://doi.org/10.1111/bcp.13718.
- Hu, A.; Keresztes, I.; MacMillan, S.N.; Yang, Y.; Ding, E.; Zipfel, W.R.; DiStasio, R.A.Jr.; Babich, J.W.; Wilson, J.J. Oxyaapa: A Picolinate-Based Ligand with Five Oxygen Donors That Strongly Chelates Lanthanides. Chem. 2020, 59, 5116–5132. https://doi.org/10.1021/acs.inorgchem.0c00372.
- Blomqvist, L.; Nordberg, G.F.; Nurchi, V.M.; Aaseth, J.O. Gadolinium in Medical Imaging–Usefulness, Toxic Reactions and Possible Countermeasures–A Review. Biomolecules 2022, 12. https://doi.org/10.3390/biom12060742.
- Sembo-Backonly, B.S.; Estour, F.; Gouhier, G. Cyclodextrins: Promising Scaffolds for MRI Contrast Agents. RSC Adv. 2021, 11, 29762–29785. https://doi.org/10.1039/D1RA04084G.
- Maksimov, A.; Kutyrev, G. Functionalized Hyperbranched Aliphatic Polyester Polyols: Synthesis, Properties and Applications. Chinese J. Polym. Sci. 2022, 1–19. https://doi.org/10.1007/s10118-022-2823-0.
- Janicki, R.; Mondry, A.; Starynowicz, P. Carboxylates of Rare Earth Elements. Chem. Rev. 2017, 340, 98–133. https://doi.org/10.1016/j.ccr.2016.12.001.
In this work, the authors propose to use hyperbranched polymers with 3D archi-tecture and high density of peripheral functional groups as a polymer platform for the creation and stabilization of gadolinium and dysprosium complexes. Boltorn H series hyperbranched polyester with terminal OH groups was chosen as the object of study. The use of the polyether functionalization approach by fragments of unsaturated carboxylic acids will increase the specificity of complexation of the hyperbranched macroligand with Gd(III) and Dy(III) ions and obtain stable complexes with good target characteristics.
In this manuscript, the authors present the intermediate results of research in project on developing approaches to the synthesis, studying the structure and photophysical properties of new Gd(III) and Dy(III) ions with hyperbranched polymers with end-fumarate and maleate groups.
In accordance with the above, the authors have adjusted the purpose of the manuscript: «In this connection, the aim of this work is to synthesize, study the structure and photophysical properties of a new complexes Gd(III) and Dy(III) ions with second-generation hyperbranched polyester with end - fumarate and maleate groups.»
Appropriate сorrections have been made to the "Introduction" section (in green).
- In addition, authors reported much higher luminescence enhancement for complex of hyberbranched polyester with Dy (III) as compared to complex with Gd (III)? What is the reason for that?
Authors comments:
At this stage of the work, it can be assumed that a significant difference in the emission intensity of the dysprosium complex compared to the gadolinium complex is apparently due to the fact that in Gd(III)ione the first excited state 6P7/2 is too far away to receive energy from triplet excited state of the ligand. While for the complex with the Dy(III) ion, the antenna effect of the ligand contributes to the efficient transfer of the absorbed energy to the Dy(III) ion, increasing the dysprosium fluorescence intensity. This assumption is presented in the “Conclusions” section: “The significant difference in the emission intensity of the complex with Dy(III) ions (4) as compared with complex with Gd(III) ions (3) (approximately 18-times) appears to be due to the fact that in Gd(III) the first excited state 6P7/2 Gd(III) is too far away to accept the energy on the triplet excited state of the ligand [52,53]. The studies will be continued to clarify the possibility of using these complexes as contrast agents in MRI.”
- Zhang, L.; Ji, Y.; Xu, X.; Liu, Z.; Tang, J. Synthesis, Structure and Luminescence Properties of a Series of Dinuclear LnIII Com-plexes (Ln=Gd, Tb, Dy, Ho, Er). J Lumin 2012, 132, 1906–1909. https://doi.org/10.1016/j.jlumin.2012.03.040.
- Bazhina, E.S.; Bovkunova, A.A.; Medved’ko, A. v; Varaksina, E.A.; Taidakov, I. v; Efimov, N.N.; Kiskin, M.A.; Eremenko, I.L. Lanthanide(III) (Eu, Gd, Tb, Dy) Complexes Derived from 4-(Pyridin-2-Yl)Methyleneamino-1,2,4-Triazole: Crystal Structure, Magnetic Properties, and Photoluminescence. Chem Asian J. 2018, 13, 2060–2068. https://doi.org/10.1002/asia.201800511.
All comments of the Reviewer are taken into account. Appropriate changes have been made to the manuscript. The authors thank the Reviewer for a detailed review of the work. The comments made are very helpful in moving our project forward in the future.

Round 2
Reviewer 1 Report
Authors make proper corrections according to reviewer remarks and I suggest to publish the paper as it is.